# Effect of Expanded Graphite on the Reaction Sintering of Boron Carbide

**DOI:** 10.3390/ma15041500

**Published:** 2022-02-17

**Authors:** Agnieszka Gubernat, Kamil Kornaus, Radosław Lach, Dariusz Zientara, Patryk Dyl

**Affiliations:** Faculty of Materials Science and Ceramics, Department of Ceramics and Refractories, AGH University of Science and Technology, Mickiewicza Av. 30, 30-059 Kraków, Poland; gubernat@agh.edu.pl (A.G.); radek.lach@poczta.fm (R.L.); zientara@agh.edu.pl (D.Z.); patryk0dyl@gmail.com (P.D.)

**Keywords:** reaction sintering, boron carbide, expanded graphite, hot-pressing

## Abstract

This paper presents novel results of research focused on reaction sintering of a mixture of expanded graphite and amorphous boron. It has been shown that as a result of combining the synthesis from the elements with sintering under pressure, dense boron carbide polycrystals (95% TD) can be obtained in which stable structures dominate, i.e., boron carbides of stoichiometry B_13_C_2_ and B_4_C. Sintering was carried out on boron excess systems, and reaction mixtures with the following mass ratios (B:C = 5:1; 10:1; and 15:1) were used. Boron excess systems were used due to the presence of additional carbon during sintering since the matrix, reactor lining, and heating elements were made of graphite. 1850 °C was considered to be the optimum reaction sintering temperature for all of the systems tested. This shows that a reduction in the sintering temperature of 200–300 °C was observed with respect to traditional sintering techniques. Micro-cracks are present in the sinters, the presence of which is most likely due to the difficulty in removing the gaseous products which accompany the boron carbide synthesis reaction. The elimination of these defects of sintering requires further research.

## 1. Introduction

Boron carbide is classified as a so-called covalent construction material. The strong B-B and B-C covalent bonding is the determining factor in the boron carbide structure. Boron carbide with a rhombohedral structure shows a wide range of phase homogeneity from B_10.4_C (8.8% at. C) to B_4_C (20% at. C) [1,2,3,4,5]. The carbon content of boron carbide significantly affects its properties and structure. Thus, knowing the B/C ratio is crucial in the research devoted to this material. The lower limit of the boron carbide homogeneity range is difficult to specify [3,5]. Figure 1 shows a B-C phase diagram with a homogeneity range of 8.8–20% at. C [1,4,5,6,7,8,9] and a diagram including low-temperature phases (i.e., boron rich boron carbides) [6]. Due to its attractive properties, it can be used in many advanced industries, which may include nuclear power, military industry, or medicine [1,4,5,10,11,12,13,14,15,16,17]. A number of distinguishing properties of boron carbide include very high hardness (~30 GPa), low density (2.52 g/cm^3^), high melting point (carbide with a B_13_C_2_ formula melts congruently at about 2450 °C), high Young’s modulus (about 450 GPa), chemical inertness, and also high neutron capture cross section (600–700 barn) [1,4,5,18].

Boron carbide can be used either as powders, forms (polycrystals) or thin films. The production of dense boron carbide polycrystals is very difficult. The main reason for the difficult sinterability of boron carbide is the existence of a strong covalent bond in the structure, which in turn is responsible for the high melting point of this compound and low values of the intrinsic diffusion coefficients [2,5,19]. An additional reason for the difficult sinterability of boron carbide is the presence of passivating oxide layers on the carbide particles, due to which inefficient mass transport mechanisms for densification occur through the gas phase. Thus, the occurrence of efficient mass transport mechanisms is limited [2,5]. The difficult sinterability can be reduced by physical activation of sintering, including the application of different sintering techniques ranging from pressureless sintering [19,20,21,22,23,24,25], HPHT (high pressure high temperature) sintering [26], hot isostatic pressing [25,27], microwave assisted sintering [28], and the currently popular electric field of assisted sintering SPS [29,30,31,32,33]. Another way to improve sinterability is to introduce sintering additives (activators). In the case of boron carbide, different sintering activators are used, which can include elements: C, Mg, Al, V, Cr, Fe, Co, Ni, Cu, Si, Ti or the compounds BN, MgO, Al_2_O_3_, Fe_2_O_3_, MgF_2_, AlF_3_ [2,5]. Carbon is the most effective additive to activate the sintering of boron carbide [2,19,20,21,22]. It effectively reduces the oxide layer formed on B4C particles, improving its sinterability by effectively activating the efficient mass transport mechanisms and hindering grain growth in polycrystals. Few attempts can be observed which have tried to combine the boron carbide synthesis reaction with sintering (in other words, with efforts at reaction sintering) [30,34,35,36]. In works devoted to the reaction sintering of boron carbide, one can find information that it is possible to produce sinter with high density [30,34,35] and desired electrical properties [36]. The authors of these works used graphite, a low-reactivity variety of carbon, as a carbon substrate, while the use of amorphous carbon or defected graphite (e.g., expanded graphite) should increase the reactivity, which should also have a beneficial effect on sintering.

The presented work therefore represents a new approach in research on reactive sintering of boron carbide, if only due to the use of a reactive form of carbon (such as expanded graphite). Furthermore, it demonstrates the possibility of producing dense sinter by reactive sintering under pressure at temperatures lower than about 200 °C compared to reactive sintering of previously synthesized powders. In addition, the degree of rearrangement is already at the lowest reaction sintering temperature (1650 °C) very high and amounts for the stable structures (i.e., B_4_C and B_13_C_2_) to more than 80%. It should also be added that the remaining 20% are boron carbides rich in boron and there are no unreacted substrates in the sample.

## 2. Materials and Methods

In order to perform the reaction sintering, the substrate mixtures with the following mass ratios (B:C = 5:1, 10:1; 15:1) were prepared. All mixtures were made with an excess of boron. It was assumed that the system would self-regulate the amount of carbon, especially since the whole process will take place in the presence of graphite (i.e., the dies used for hot pressing are made of graphite, the lining and heating elements are also made of graphite). The following items were used as substrates of the synthesis reaction: *Fluka* amorphous boron powder (catalog no. 15580) of chemical purity 95–97% and average particle size 1.5–2 µm and expanded graphite obtained by exfoliation of *Sinograf* expanded graphite with catalog no EG290. The exfoliation was carried out in a reactor using microwaves with a maximum power of 1200 W/g. The exfoliation time was 60 s. After exfoliation, the specific surface area of graphite, as measured by BET, was 38.25 ± 0.55 m^2^/g. Figure 2 shows the SEM morphologies of the substrates used for the boron carbide synthesis reaction.

After weighing, the substrates were wet homogenized in a ball mill in ethanol with SiC spherical grinders for 12 h. Then, after evaporation of the alcohol, portions of the mixtures were weighed for hot pressing. Hot pressing was carried out at 1650, 1750, 1850, and 1900 °C in a flow of inert gas (i.e., argon). The samples were kept at each temperature for one hour. A constant pressure of 25 MPa was applied during the process. The temperature progression was 10 °C/min. Sintering under pressure was carried out in a Thermal-Technology high-pressure sintering press. The phase composition of the powders and sinter was determined using an X-ray diffractometer from PANalytical (model Empyrean). The measurements were made using monochromatic radiation with a wavelength corresponding to the K emission line of copper, in the angular range 5–90° on a 2θ scale, and the step of the goniometer was 0.008°. Qualitative analysis of the phase composition was performed using X’Pert HighScore Plus version 3.0e computer program developed by PANalytical. To identify the phase composition, the obtained diffractograms were compared with the FIZ Karlsruhe 2012 powder diffraction database and the PDF-2 database (2004). The polycrystals were subjected to apparent density tests using the Archimedes method. Since the phase composition analyses revealed that the polycrystals obtained consisted of boron carbides with slightly different B:C stoichiometry, the relative density of the polycrystals was also calculated. A theoretical boron carbide density of 2.52 g/cm^3^ was used to calculate the relative density [1,4,5]. The sinter with the highest density was ground and polished to produce metallographic specimens. Then, the specimens were chemically etched in order to visualize the elements characterizing the microstructure in molten alkaline salts of 25 wt.% KNO_3_ and 75 wt.% KOH at 450 °C. The metallographic sections were used to observe the microstructure on the Nova Nano SEM 200 FEI Company scanning microscope.

## 3. Results

### 3.1. X-ray Diffraction Anallysis

Figure 3 shows the results of phase composition analyses of polycrystals made from mixtures with different B:C mass ratios, sintered at different temperatures. The phase composition measurements show that under the applied reaction sintering conditions, each of the prepared samples achieved either a high or complete degree of conversion. Boron carbides of different stoichiometry are present in the phase composition of the samples. The rhombohedral boron carbide exhibits a wide range of phase homogeneity from the carbide with stoichiometry B_10_C to the carbide with stoichiometry B_4_C, with the most stable structures being those with high carbon content (i.e., the carbides with stoichiometry B_13_C_2_ and B_4_C). The systems in reaction sintering tend to become more and more carbon saturated in the carbide structure (i.e., the stoichiometry described by the patterns B_13_C_2_ and B_4_C) [3]. This regularity is also observed in the case of the investigated materials. The increase in crystallinity of the carbide structure can be seen in the X-ray diffractograms presented in Figure 3. For all studied systems, regardless of the B:C mass ratio, as the reaction sintering temperature increases, the reflections in the X-ray diffractograms become sharper and more intense (Figure 3). At the lowest sintering temperature of 1650 °C, low-temperature phases are present (i.e., boron carbide with a tetragonal structure, the so-called boron-rich boron carbide B_48_(B_2_C_2_) = B_50_C_2_ (Table 1)). This boron carbide structure is present in the samples made from the mixtures with B:C mass ratio = 10:1 and 15:1. In the sample made from the mixture with the lowest boron to carbon excess, only boron carbides with the highest carbon saturation of the structure were identified already at 1650 °C. At 1750 °C, only the carbides with the most stable structures (i.e., B_13_C_2_ and B_4_C) are present. An increase of temperature by another 100 °C does not change the phase composition with respect to the samples made from the mixtures with B:C = 10:1 and 15:1. In the case of a sample made from the mixture with B:C = 5:1 phase ratio, the system reaches maximum carbon saturation, which is manifested by the presence of graphite in the sinter (Table 1). At the highest reaction sintering temperature (i.e., 1900 °C), graphite also appears in the sinter made from the mixtures with B:C = 10:1 and 15:1 (Table 1). A correlation between temperature and carbon saturation of the boron carbide structure can be observed. As the amount of carbon in the boron carbide rhombohedral structure increases, the values of the lattice parameters of the elementary cell decrease and thus the size of the elementary cell decreases, which is manifested by a shift of the main reflections for angles ~35 (104) and 38° (021) 2 theta towards larger values of angle 2 theta (Figure 4) [35,37]. This regularity can be observed for the studied samples.

### 3.2. Density of Polycrystals

The presented measurements (Table 2; Figure 5) indicate that it is possible to combine the synthesis of boron carbide from elements with the simultaneous sintering under pressure. The produced polycrystals already at the lowest reaction sintering temperature show a high relative density close to 70% (Figure 5). The increase in sintering temperature is manifested in all samples by an increase in density, regardless of the initial composition of the samples. The highest density is obtained at a reaction sintering temperature of 1850 °C (Figure 5). Polycrystals made from the mixtures with B:C = 5:1 and 10:1 mass ratios show high relative density, close to 96%, while the sinter prepared from a mixture with B:C = 15:1 mass ratio reaches a relative density of 93%. Increasing the sintering temperature up to 1900 °C results in a slight decrease in density for the sinter made from the mixtures with B:C = 5:1 and 10:1, while a decrease in density of about 2% was observed for the sample made from the mixture with the highest boron to carbon (graphite) ratio. This slight decrease in density can be related to the free carbon present in the sinter, in the form of graphite, which was identified by XRD measurements.

### 3.3. Microstructure Analysis

Figure 6 illustrates an example of the microstructure of a non-chemically etched sinter. The picture shows defects in the form of pores. Indeed, the majority of the picture shows a uniform degree of grey, which gives grounds to suggest that the obtained polycrystals are homogeneous in terms of chemical composition. It is also very likely that they are single phase.

The microstructure analysis was performed on the highest density polycrystals (i.e., the ones sintered at 1850 and 1900 °C). The prepared metallographic specimens were chemically etched to visualize the microstructural elements of the polycrystals. Figure 7 shows SEM images of samples with different B:C initial compositions, sintered at 1850 and 1900 °C. It is evident from the images that the produced polycrystals show high density. The microstructure of the analyzed polycrystals also shows pores (i.e., the darkest objects and micro-cracks). SEM micrographic observations also show that a significant number of grains are twinned crystals [38], which constitute a characteristic element of the boron carbide microstructure. This type of close formation is called a polysynthetic close formation and parallel adhesion planes are present.

Increasing the sintering temperature results in grain growth, which is characteristic for sintering. Regardless of the initial composition, all polycrystals sintered at a temperature of 1850 °C are characterized by a microstructure typical for high-density sinters (Figure 7a–c). Increasing the temperature to 1900 °C, as shown by the apparent density measurements (Figure 6), results in a slight decrease in the density of the sinters, which is not noticeable in their microstructure (Figure 7d–f). The density of the polycrystals may be adversely affected by the graphite present in them, identified by XRD analysis (Figure 3 and Table 1). Trace amounts of free carbon, most likely graphite, were also identified by the EDS analysis (Figure 8).

In the following Figure 9, SEM microstructure images of polycrystals sintered at 1900 °C, differing in the initial composition (i.e., B:C mass ratio) are summarized. It is observed that the larger the initial B:C mass ratio was, the higher the grains are in the sinter.

## 4. Discussion

Reaction sintering, in which consolidation occurs by means of a reaction between substrates, has been successfully implemented for binary oxide systems and binary metal systems [39]. The underlying assumption of this method is that the molar volume of the resulting product has a larger molar volume as compared to the substrates. In the present case, the molar volume of the product (for B_4_C is 22.2 cm^3^/mol, for B_13_C_2_ is 66 cm^3^/mol) is similar to the substrates. Assuming boron carbide B4C stoichiometry, the molar volume of the substrates ranges from 24 to 27 cm^3^/mol, while for boron carbide B_13_C_2_ stoichiometry it ranges from 71 to 81 cm^3^/mol. However, it should be taken into account that boron carbides with the same structure but different stoichiometry are formed during sintering, which affects the molar volume of the product. Therefore, it can be thought that there are no contraindications to carry out reaction sintering of boron carbide. In this work, it has been shown that it is possible to combine the synthesis and sintering of a mixture of amorphous boron and expanded graphite, and thereby obtain the dense sintered boron carbide. It was found that all of the samples with different starting compositions (with different boron to carbon mass ratios), produced by sintering under pressure, exhibited a high degree of conversion (Figure 3 and Table 1). At the lowest reaction sintering temperature, boron carbide with a rhombohedral structure and stoichiometry B_13_C_2_ dominates in the samples, and tetragonal structure boron-rich carbide with stoichiometry B_48_B_2_C_2_(B_50_C_2_) can also be identified. Increasing the sintering temperature results in a higher carbon saturation of the boron carbide structure and the formation of a carbide with a stable structure, mainly B_13_C_2_, and a carbide with rhombohedral structure and the highest carbon atomic proportion with stoichiometry B_4_C (Table 1). The further increase of the sintering temperature results in maximum carbon saturation of the structure, so there is excess carbon (i.e., graphite) in the phase composition of both sintered materials (Figure 3 and Table 1). 1850 °C can be considered as the optimal reaction sintering temperature. Then, all polycrystals, produced from the mixtures with different mass ratios of B:C substrates, show the highest relative density, which ranges from 93 to 96%. Increasing the sintering temperature up to 1900 °C causes a slight decrease in the density of the sinter, which may be related to the presence of graphite precipitates, identified by the XRD analysis (Figure 3) and also by the EDS analysis (Figure 8). When compared to sintering by traditional methods of fine-grained boron carbide powders with stoichiometry B_13_C_2_ or B_4_C [2,5], the reaction sintering can result in 200 to 300 °C lower sintering temperature under the described conditions.

Phase composition measurements show another regularity. The higher the initial B:C ratio, the lower the carbon saturation of the carbide structure at the lowest synthesis temperature (Table 1). This observation seems to be obvious: the higher the excess of boron in relation to carbon, the systems require higher temperature, and thus longer time for carbon saturation of the structure, in other words, the formation of stable boron carbide structures. Moreover, as shown in the phase diagram (Figure 1), the formation of boron-rich structures is favored in the case of a significant excess of boron.

The opinions on the effect of reaction synthesis on densification are divided [39,40,41], but [39] argues that there is no evidence that reaction energy has a direct effect on the driving forces of sintering. Thus, the main benefit of reaction sintering consists in the elimination of the step of prior synthesis of the compound, which is particularly important for complex compounds. In the present case, the beneficial effect of the pressure applied during sintering is marked (mainly on the acceleration of sintering and higher density of sinter). All studied samples, at a much lower temperature, reach high density (Table 2 and Figure 5), as compared to the polycrystals obtained by traditional sintering techniques [19,20,21,22,23,24]. The results presented are in agreement with the description of reaction sintering [39], according to which it is unlikely that pressure affects the synthesis reaction. However, it should be emphasized that due to the applied pressure, it is possible to increase the number of contacts between substrates and thus accelerate the synthesis reaction. Moreover, the efficiency of the synthesis is influenced by the use of carbon (i.e., expanded graphite) as a substrate. This graphite, as a result of intercalation and subsequent exfoliation, acquires a specific structure and microstructure. It is a highly porous and defected form of carbon (Figure 2), and thus is highly reactive [42,43,44].

The microstructure analysis shows that the grain size in the sintered samples depends, among other things, on the initial composition of the boron-carbon mixtures (Figure 9). It was observed that the lower the B:C mass ratio, the smaller the grains in the sinter. It is likely that mass transport mechanisms, associated with the reaction and sintering, occur most slowly, in a system that is fully reacted and the structure saturated with carbon already at the lowest reaction sintering temperature (i.e., 1650 °C (Table 1)). Similar observations can be found in the work [36]. In addition, graphite was identified in the sinter, which was made from a mixture with B:C mass ratio = 5:1, by the XRD analysis (Figure 3) and carbon precipitates by the EDS analysis (Figure 8). The occurrence of graphite in the form of precipitates can also effectively limit the grain growth in the sinter.

The polycrystals produced exhibit a microstructure characteristic for rhombohedral structure of boron carbide, composed mainly of twinned grains (Figure 7). The defects in the form of pores and micro-cracks were also identified in the samples. There can be many reasons for the formation of micro-cracks. They can be formed as a result of plastic deformation during hot pressing, and can also be constituted by the encapsulation of gaseous products formed during the synthesis of boron carbide in a tight graphite matrix. Most of the works on boron carbide sintering tend to conclude that micro-cracks are formed due to thermal expansion anisotropy of differently oriented grains in the sinter [22,34,45,46]. The calculations of thermal expansion coefficients, of boron carbide with stoichiometry B_13_C_2_, depending on the crystallographic direction (as presented in [22,46]), show that there are differences in the values of these coefficients. The larger the values, the higher the temperature (Table 3).

It is also possible that micro-cracks occur due to the smaller molar volume of the product compared to the substrates. An equally possible reason is the possibility of the formation of micro-cracks due to the difficulty or even decompression of the gaseous products of the carbide synthesis reaction contained inside the samples. This cannot be excluded. Pressureless sintering of the samples formed from the mixtures of amorphous boron and expanded graphite with different B:C mass ratios (the same as in the case of sintering under pressure) was also attempted. In each case, irrespective of the initial mixture composition and sintering temperature, the samples disintegrated after the process (Figure 10). On the other hand, however, they showed a high degree of reactivity (Figure 11). It is suggested that the disintegration occurs due to the decompression of the gaseous synthesis products contained inside. The analysis of Figure 10 further shows the favorable effect of pressure on the yield of the synthesis reaction For all of the samples sintered under pressure, at the lower temperature of 1750 °C, the stable boron carbide structures of stoichiometry B_13_C_2_ and B_4_C become the majority.

In the literature we can find residual information on the reactive sintering of boron carbide. In the work of Kalandadze et al. [34], it was found that it is possible to produce dense sinters by the hot-pressing mixed amorphous boron and carbon and crystalline boron variety and carbon. The authors of this work have shown that the combination of synthesis and sintering allows them to obtain dense sinters at the temperatures close to 1800 °C in the case of a mixture of crystalline boron and carbon. According to the authors of the discussed work, a high degree of substrate homogenization is important, since, as they claim, the synthesis reaction depends on the diffusion of boron towards carbon [34].

On the other hand, however, [46] proposes another mechanism for the synthesis of boron carbide using a specific form of carbon (i.e., expanded graphite). The work [47] proposed a mechanism for the synthesis of boron carbide using expanded graphite in a layered system. Basing on the experiments, it was concluded that the transport of carbon towards boron and thus the process of carbon incorporation into the rhombohedral structure of boron is very likely. The gaseous carriers of carbon may be residues of oxidizing agents used in the intercalation of expanded graphite and oxygen, which is an impurity of argon used during sintering. The mechanism of boron carbide synthesis can be described by the following equations:(1)$$2C+O_{2}=2C$$
(2)$$4B+2{\mathrm{CO}}_{\left(g\right)}=B_{4}C+{\mathrm{CO}}_{2\left(g\right)}$$
(3)$${\mathrm{CO}}_{2}+C↔2\mathrm{CO}$$

Probably due to its large specific surface area (30–50 m^2^/g), the expanded graphite is oxidized by the oxygen present in the system Equation (1), and then carbon in the form of carbon monoxide is transported towards boron wherein boron carbide is synthesized (according to Equation (2)). The carbon dioxide is continuously reduced to carbon monoxide according to Equation (3). From the equations presented, it can be seen that significant amounts of gaseous products are formed during the synthesis, which can lead to the formation of micro-cracks if they cannot escape freely from the graphite matrix. Moreover, when the synthesis process coexists with sintering of the system, this effect can be enhanced. In order to exclude the discussed cause, a pressurized sintering of a mixture with a B:C mass ratio was carried out without external pressure up to a synthesis temperature of 1550 °C. Only after it was exceeded, the system was hot pressed at 25 MPa.

The density measurements and microstructure analysis were then performed on the sample. Accordingly, it was found that the sample is less dense (~92% TD), and the grains are significantly smaller, but micro-cracks are still present (Figure 12b). The comparison of microphotographs of the samples with the same initial composition, sintered at the same temperature (Figure 12a,b), shows that the micro-cracks in the sample sintered without external pressure are smaller in size. This observation allows us to suggest a hypothesis that the presence of micro-cracks is related to the boron carbide synthesis, occurring simultaneously with sintering. In the course of the synthesis according to reactions, described by Equations (1)–(3), numerous gaseous products are formed, which in the case of sintered samples under pressure have a difficult evacuation path both since the sample is enclosed in a tight matrix and since the sample is compacted simultaneously with the synthesis. Running the process without external pressure up to the synthesis temperature facilitates the evacuation of the gaseous products but, on the other hand, reduces the densification of the sinter.

## 5. Conclusions

The following conclusions can be drawn from the work:It has been shown that it is possible to combine elemental synthesis and sintering of boron carbide when expanded graphite, a porous and reactive form of carbon, is used as a carbon substrate;Dense polycrystals can be obtained from the mixtures of substrates with a significant excess of boron relative to carbon from B:C of 5:1 to B:C of 15:1;Reaction sintering was carried out on all mixtures in the temperature range of 1650–1900 °C. The best results considering polycrystalline density and synthesis reaction efficiency were obtained at 1850 °C. This is a significant reduction in sintering temperature, as compared to traditional techniques, of about 200–300 °C. The sinters obtained at this temperature, regardless of the starting composition, exhibit relative densities in the 90–96% TD range;At each of the applied reaction sintering temperatures, the obtained polycrystals consist only of boron carbides of different stoichiometry, dominated by boron carbides with stable structures (i.e., B_13_C_2_ and B_4_C). At the lowest reaction sintering temperature of 1650 °C, a boron-rich boron carbide with stoichiometry B_50_C_2_ can be identified. On the other hand, at the highest reaction sintering temperature, graphite precipitates appear in the polycrystals due to the maximum carbon saturation of the boron carbide structure;The micro-cracks, visible in the images of microstructures, are most likely formed due to the difficulty in the free discharge of the gaseous products of carbide synthesis. When the sample is in the graphite matrix, it is subjected to external pressure and begins to sinter. Separating the carbide synthesis from sintering reduces this effect but worsens the density of the materials produced.

## Figures and Tables

**Figure 1 materials-15-01500-f001:**
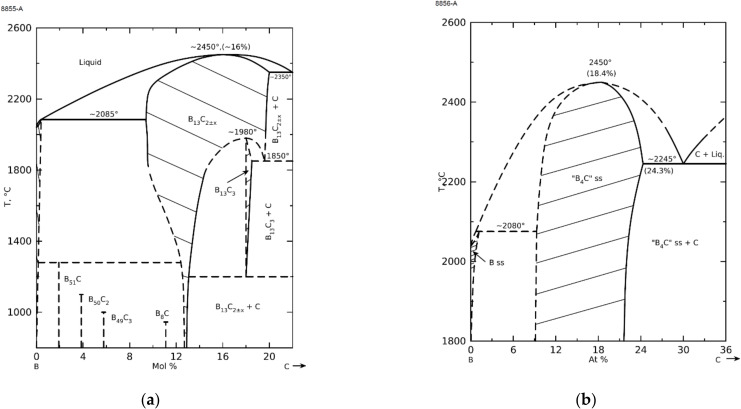
Boron—carbon phase diagrams according to (**a**) Ekbom and Amundin [6] and (**b**) according to Beauvy [9] (Phase Equilibria Diagrams PC Database (NIST Standard Reference Database 31), Version 4.0, The American Ceramic Society and the National Institute of Standards and Technology, 2013. Figure Numbers 8855-A, 8856-A).

**Figure 2 materials-15-01500-f002:**
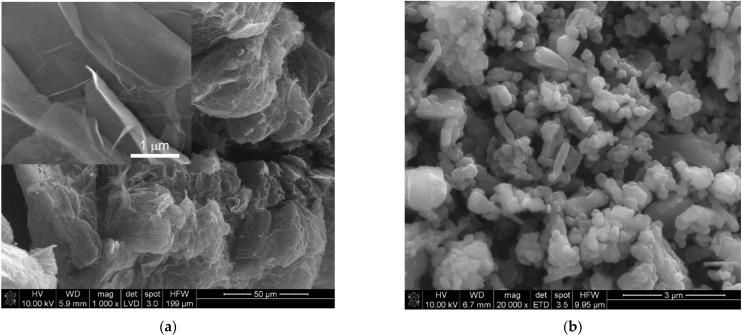
Substrate morphologies: expanded graphite (**a**) and amorphous boron (**b**).

**Figure 3 materials-15-01500-f003:**
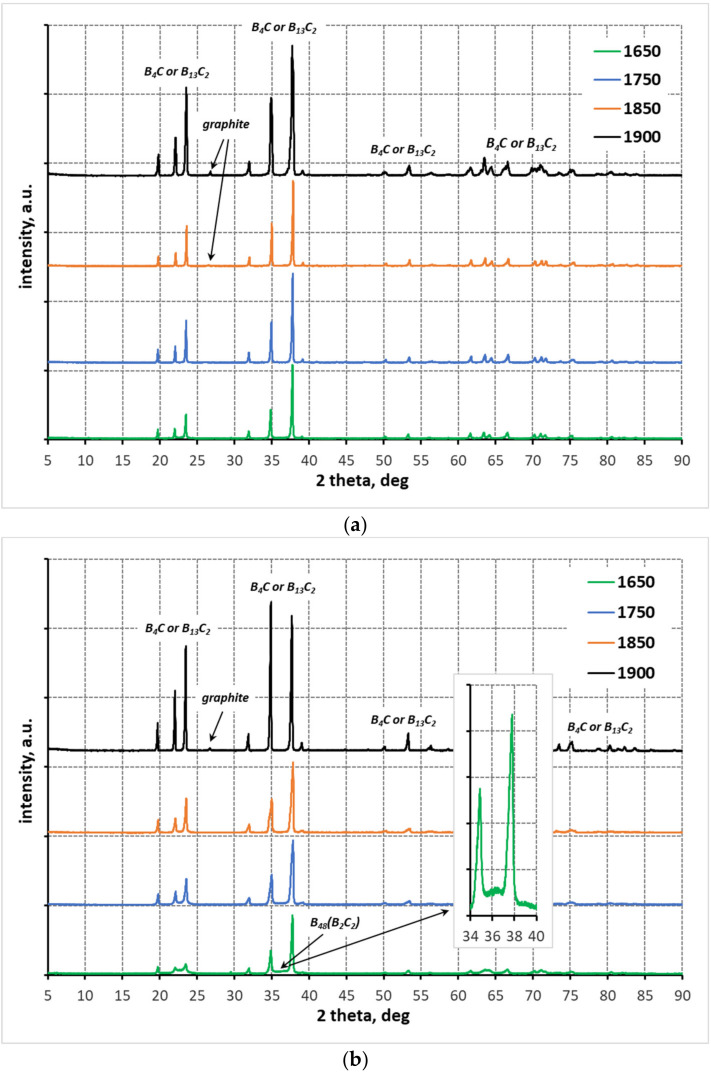
X-ray diffractograms of sintered polycrystals at different temperatures, made from mixtures with different B:C mass ratios; (**a**) 5:1 (**b**) 10:1 (**c**) 15:1 (the X-ray diffractograms show reflections coming from graphite and tetragonal B_48_(B_2_C_2_). Other reflections are characteristic of rhombohedral boron carbides with different stoichiometry).

**Figure 4 materials-15-01500-f004:**
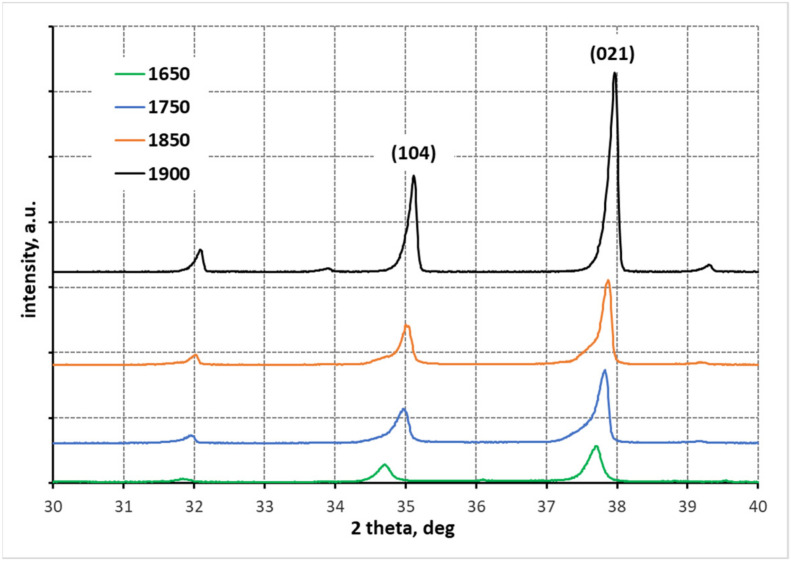
Fragment of X-ray diffractograms from 30 to 40° 2 theta (samples with B:C = 5:1).

**Figure 5 materials-15-01500-f005:**
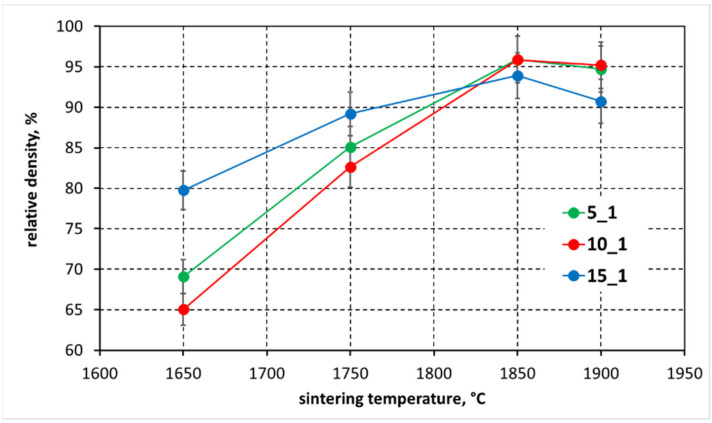
Results of sinters relative density measurements as a function of reaction sintering temperature and initial composition of samples. The 5_1, 10_1 and 15_1 descriptions in the graph refer to the B to C molar ratios used in the reaction mixtures which are B:C = 5:1, 10:1 and 15:1 respectively.

**Figure 6 materials-15-01500-f006:**
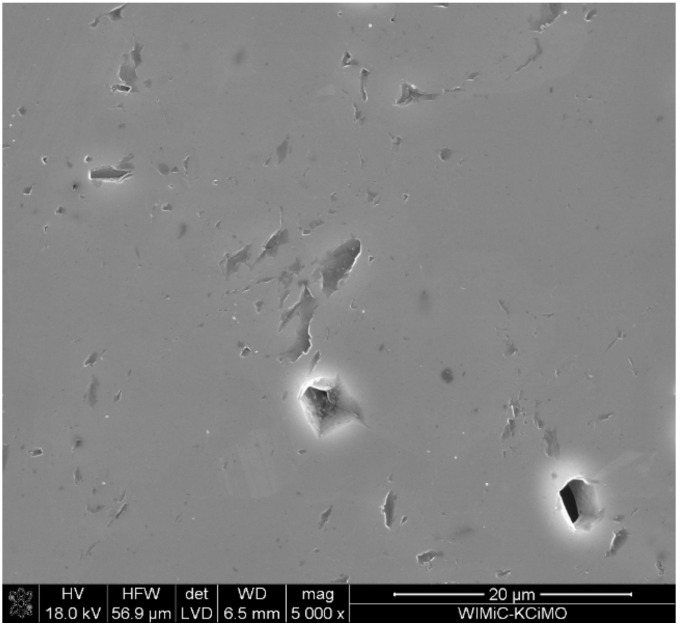
SEM image of a boron carbide sinter (B:C = 10:1, T_rs_ = 1850 °C) that was not chemically etched.

**Figure 7 materials-15-01500-f007:**
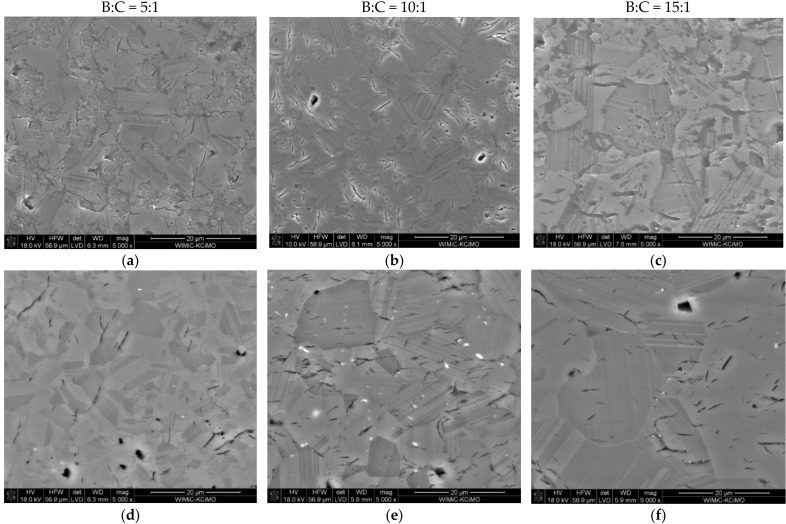
SEM images of boron carbide polycrystals, with different starting compositions, sintered at: 1850 (**a**–**c**) and 1900 °C (**d**–**f**).

**Figure 8 materials-15-01500-f008:**
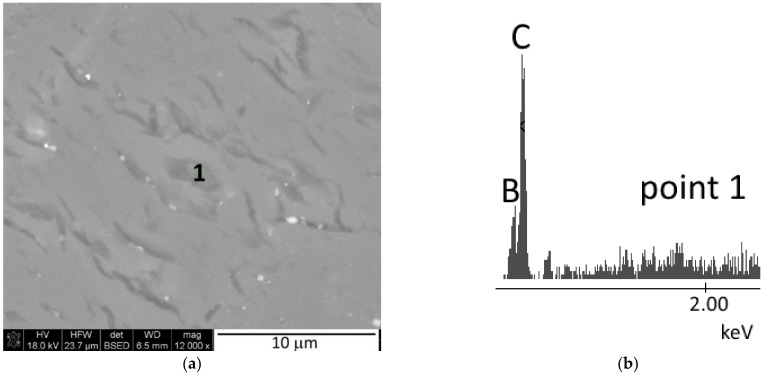
EDS analysis (**b**) of the micro-area marked as point 1 (**a**) of the sample obtained from a mixture with B:C = 15:1 ratio, sintered at 1900 °C.

**Figure 9 materials-15-01500-f009:**
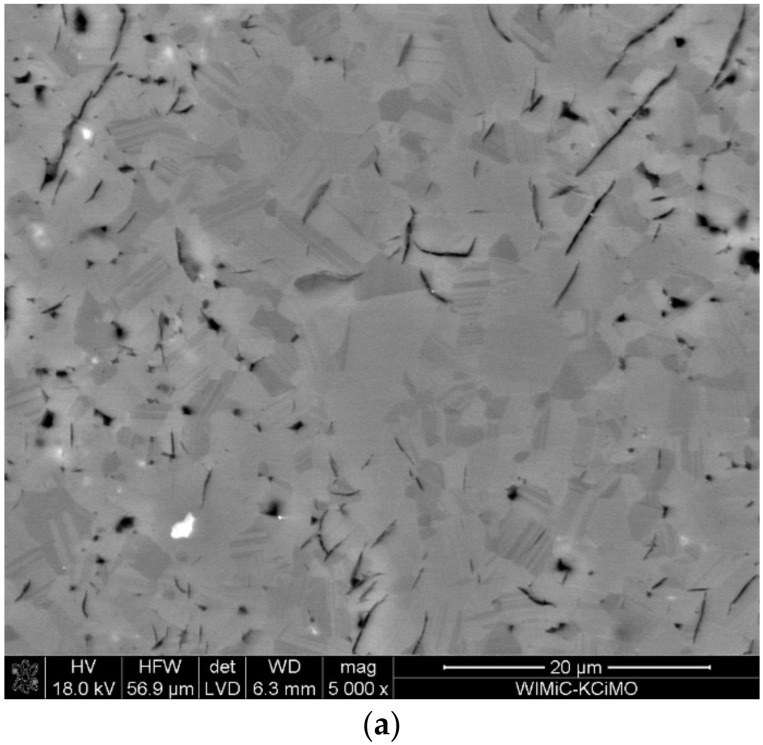
SEM micrographs of sinters made from mixtures with different starting compositions that were reaction sintered at 1900 °C: (**a**) B:C = 5:1; (**b**) B:C = 10:1; and (**c**) B:C = 15:1.

**Figure 10 materials-15-01500-f010:**
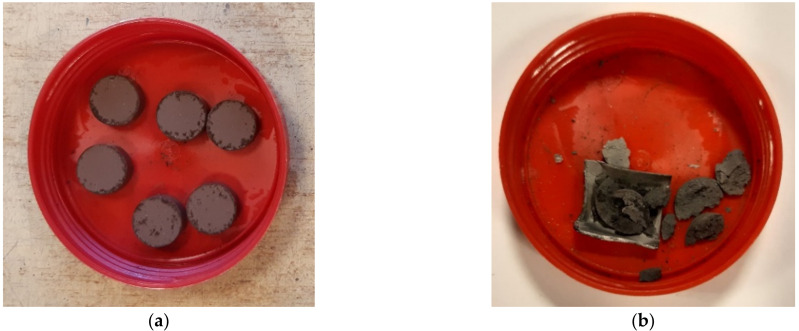
Samples prepared for pressureless sintering (**a**) and sample after pressureless sintering process (**b**).

**Figure 11 materials-15-01500-f011:**
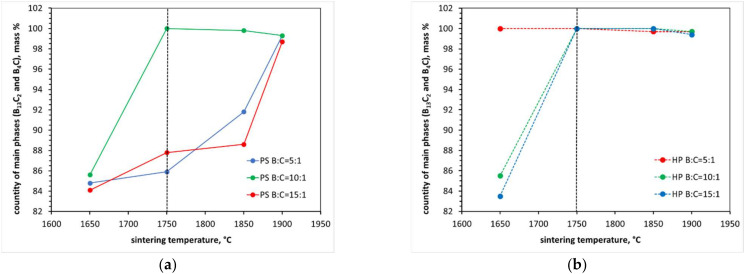
Measurement results of quantitative phase composition including stable boron carbide structures, i.e., B_13_C_2_ and B_4_C, as a function of reaction sintering temperature and sintering method used: PS, pressureless sintering (**a**); HP, hot pressing (**b**).

**Figure 12 materials-15-01500-f012:**
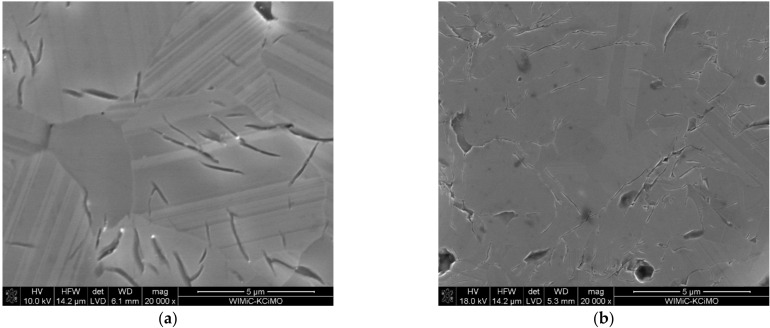
SEM microstructures of samples with B:C = 10:1 composition: (**a**) sintered under 25 MPa pressure to a final temperature of 1850 °C and (**b**) sintered without external pressure to 1550 °C and then under 25 MPa pressure to 1850 °C.

**Table 1 materials-15-01500-t001:** Phase composition of samples prepared from mixtures with different B:C mass ratios, as a function of reaction sintering temperature.

B:C Ratio	5:1			10:1			15:1		
	Phase Composition, Mass %								
Sintering Temperature, °C	B_13_C_2_ B_4_C	B_48_(B_2_C_2_)	C	B_13_C_2_ B_4_C	B_48_(B_2_C_2_)	C	B_13_C_2_ B_4_C	B_48_(B_2_C_2_)	C
1650	100	-	-	85.5	14.5	-	83.5	16.5	-
1750	100	-	-	100	-	-	100	-	-
1850	99.7	-	0.3	100	-	-	100	-	-
1900	99.7	-	0.6	99.7	-	0.3	99.4	-	0.6

**Table 2 materials-15-01500-t002:** Results of measurements of apparent and relative densities of boron carbide sinters obtained by reaction sintering of mixtures with different B:C mass ratios using the pressure assisted sintering method.

Mass Ratio B:C	5:1		10:1		15:1	
Temperature, °C	Apparent Density, g/cm^3^	Relative Density, %	Apparent Density, g/cm^3^	Relative Density, %	Apparent Density, g/cm^3^	Relative Density, %
1650	1.74	69.08	1.64	65.04	2.01	79.76
1750	2.14	85.11	2.08	82.63	2.25	89.21
1850	2.42	95.89	2.42	95.87	2.37	93.90
1900	2.39	94.71	2.40	95.19	2.18	90.73

**Table 3 materials-15-01500-t003:** Values of coefficients of thermal expansion, according to works [45,46].

	TEC (α∙10^–6^ K^–1^) of Boron Carbide B_13_C_2_		
Temperature, °C	α_a_	α_c_	α_av_
20–435 [46]	4.76	4.40	4.64
20–600 [45]	5.29	6.25	5.61
435–700 [46]	6.40	5.96	6.25

## Data Availability

Data will be shared via email upon request from the reader.

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
