# Peer review of "Effect of Expanded Graphite on the Reaction Sintering of Boron Carbide"

_materials, 2022, doi:10.3390/ma15041500_

Round 1
Reviewer 1 Report
The manuscript reports interesting experiments on the preparation of boron carbide ceramic sinter bodies from expandable graphite as starting material. The text is well written; maybe slightly lengthy due to some redundancy; and perhaps a few of the SEM pictures could have been omitted. But nevertheless, the manuscript is for sure worth to be published. Just some comments on details:
- line 58 and reference [45]: use indices for the chemical formulae.
- line 60ff: 3 times "attempts" - improve!
- line 94ff: Which device was used for hot pressing?
- line 98ff: Which device was used for XRD measurements, and which software for the Rietfeld analysis?
- line 108: "salts of 25% KNO3 and 75% KNO3" ???
- line 116ff, and caption to Fig. 3: This are X-ray diffractograms, but no "images", and not yet "phase composition analysis". It would be very helpful if you could label at least some of the peaks with the corresponding phase (which boron carbide?), and possibly with the indices of the X-ray reflection.
- The word "roentgenogram" is in use practically only in the eastern part of europe, until Germany, where Wilhelm Conrad Röntgen lived and worked. Better to use "X-ray diffractogram". (Röntgen himself was using the term "X-Strahlen".)
- Fig. 4: (104) and (021) from which boron carbide phase? (Put it to the figure, or to the caption).
- line 192: Polysythetic twinning is a common feature as well as for minerals like axinite. It seems to have a positive impact on the mechanical properties of boron carbide, see, e.g., Qi An et al., Superstrength through Nanotwinning, Nano Letters 2016 16 (12), 7573-7579. DOI: 10.1021/acs.nanolett.6b03414. You could consider to mention this. Or give other references on this topic.
- Below Fig. 7: "... results in the grain growth..."
- Discussion: I cannot follow the arguments on the molar volume. If you discuss B4C, then (from my point of view) you should use 4 times the molar volume of boron + 1 times for graphite.
- Table 3: You give data for the rhombohedral phase B13C2 from the literature, and then indeed you have 2 coefficients αa and αc. But in this reference there are found informations also for boron carbide phases with similar composition, but lower (monoclinic) symmetry. These would require 4 coefficients in total. (Which are not given in this reference.) You should at least mention that also your B13C2 is rhombohedral. And another point: the unit for α is 10-6 K-1. Thus,
"α/10-6 K-1" would be correct. - Chemical equation (1): 2 C + O2 ↔ 2 CO (actually, two stacked arrows, from left to right & from right to left would be even better, but I cannot produce this here).
Anyway, a fine paper!
Author Response
Dear Reviewer,
Thank you very much for your review. All comments of an editorial nature, language errors, and additions to the measurement methodology and literature have been taken into account and corrected in the text.
We would like to respond to some of the comments.
Comment No. 8. Fig. 4: (104) and (021) from which boron carbide phase? (Put it to the figure, or to the caption).
Boron carbide with rhombohedral structure shows a wide range of phase homogeneity from B10C to B4C. Thus, the B:C ratio is different in this case, which manifests itself in a different distribution of carbon and boron atoms in the carbide structure. The increase in carbon saturation leading to the formation of a rhombohedral structure with B4C stoichiometry is visible in the X-ray diffractograms in the form of a shift of the main reflections (104) and (021) towards larger values of angle 2 theta. In the text we have introduced the addition that it is boron carbide with rhombohedral structure.
Comment No. 11. Discussion: I cannot follow the arguments on the molar volume. If you discuss B4C, then (from my point of view) you should use 4 times the molar volume of boron + 1 times for graphite.
Thank you for your very valid comment. We have recalculated the molar volumes and corrected the calculations.
Comment No. 12. Table 3: You give data for the rhombohedral phase B13C2 from the literature, and then indeed you have 2 coefficients αa and αc. But in this reference there are found informations also for boron carbide phases with similar composition, but lower (monoclinic) symmetry. These would require 4 coefficients in total. (Which are not given in this reference.) You should at least mention that also your B13C2 is rhombohedral. And another point: the unit for α is 10-6 K-1. Thus, "α/10-6 K-1" would be correct.
In our work we obtained mainly boron carbide with rhombohedral structure by reaction sintering. Boron carbide with monohedral structure described in the cited work is extremely rare. As you suggested, we have underlined in the text that we refer to rhombohedral boron carbide.
Thank you again for all your comments.
Please see the attachment.

Reviewer 2 Report
In this manuscript, the authors sintered boron carbide polycrystals using a mixture of graphite and amorphous boron. The influences of boron to carbon mass ratio and sintering temperature were studied. XRD and SEM were employed to exam to the crystal structure and surface morphology of sintered materials. Here are the comments and questions:
- I would suggest the authors label the peaks of XRD patterns in Figure 3. What is the meaning of each peak?
- In Figure 4, the figure legend shows the spectra are collected for samples with B:C 5:1 sintered at 1650C, 1750C, 1850C, and 1900C, respectively. However, the figure caption indicates that they are for samples with B:C 10:1 sintered in 1850C. Which one is correct?
- Figure 7c has a different magnification compared to Figure 7a and 7b. Is that possible to use the same magnification for better comparison?
- Figure 9 a-c are exactly the same as Figure 7 d-f. Please either delete Figure 9 or use new figures.
Author Response
Dear Reviewer,
Thank you very much for your review. We have taken all comments into account and made corrections. We have described the reflections in the X-ray difractograms, correctly captioned Fig. 4, standardized the magnifications of the SEM microphotographs in Fig. 7, and replaced the microphotographs in Fig. 9.
Round 2
Reviewer 2 Report
The authors have responded to most of my comments properly. One question is still remaining regarding the XRD patterns in Figure 3. What do these peaks higher than 60 deg represent? They should be either labeled in the figure or explained in the main text.
Author Response
Thank you very much for your comment. We have added the missing description of the phases present in the diffractogram. All previously undescribed peaks correspond to rhombohedral boron carbide phases of different stoichiometry. I have also corrected the description of Figure 3. Is the current description of the diffractogram sufficient?